# Neuroprotective Effects of gH625-lipoPACAP in an In Vitro Fluid Dynamic Model of Parkinson’s Disease

**DOI:** 10.3390/biomedicines10102644

**Published:** 2022-10-20

**Authors:** Teresa Barra, Annarita Falanga, Rosa Bellavita, Jessica Pisano, Vincenza Laforgia, Marina Prisco, Stefania Galdiero, Salvatore Valiante

**Affiliations:** 1Department of Biology, University of Naples Federico II, 80125 Naples, Italy; 2Department of Agricultural Sciences, University of Naples Federico II, 80055 Portici, Italy; 3Department of Pharmacy, School of Medicine, University of Naples Federico II, 80131 Naples, Italy

**Keywords:** Parkinson’s disease, PACAP, millifluidic model

## Abstract

Parkinson’s disease (PD) is an aggressive and devastating age-related disorder. Although the causes are still unclear, several factors, including genetic and environmental, are involved. Except for symptomatic drugs, there are not, to date, any real cures for PD. For this purpose, it is necessary develop a model to better study this disease. Neuroblastoma cell line, SH-SY5Y, differentiated with retinoic acid represents a good in vitro model to explore PD, since it maintains growth cells to differentiated neurons. In the present study, SH-SY5Y cells were treated with 1-methyl-4-phenylpyridinium (MPP^+^), a neurotoxin that induces Parkinsonism, and the neuroprotective effects of pituitary adenylate cyclase-activating polypeptide (PACAP), delivered by functionalized liposomes in a blood–brain barrier fluid dynamic model, were evaluated. We demonstrated PACAP neuroprotective effects when delivered by gH625-liposome on MPP^+^-damaged SH-SY5Y spheroids.

## 1. Introduction

Parkinson’s disease (PD) is the second most common neurodegenerative disease, with a prevalence of 7 million people worldwide and a prediction of 9 million by 2030 [1,2,3]. It is a chronic, slow-progressing disorder, worsening with advancing age due to motor and non-motor symptoms [4,5]. Pathological manifestations are characterized by bradykinesia, rigidity, and tremor [6]. Loss of autonomic cognition are attributable to both abnormal protein aggregates containing α-synuclein (Lewy bodies) in regions of the central nervous system (CNS) [7] and to the death of dopaminergic neuron (DAn) in the substantia nigra [8]. The molecular mechanism underlying PD are still unknown, and thus treatments are not available. To overcome this limitation, in vitro studies can be very useful. It is, therefore, mandatory to develop a stable and reliable cell model in order to study the pathogenesis of PD to better validate adequate therapeutic agents. Neuroblastoma cell line, SH-SY5Y can represent an ideal in vitro PD cell model (67.6% and 19.7% like differentiated cells) [9]. They derived from human neuroblastoma SK-N-SH after three subclone, resulting from a biopsy of human bone marrow. This cell line has been used extensively for the expression of neuronal markers (tyrosine and dopamine-β-hydroxylase) and the specific absorption of norepinephrine, and it expresses one or more neurofilament proteins; these cells also express opioid, muscarinic and nerve growth factor receptors [10]. Therefore, they are adrenergic in the phenotype but also express dopaminergic markers and, as such, are used to study PD, neurogenesis, and other cell characteristics cerebrally [9,10,11,12,13]. SH-SY5Y cells can interact spontaneously between two in vitro phenotypes, neuroblast-like cells, and epithelial cells. The dividing cells are locked in an early neuronal differentiation stage because of few neuronal markers [14,15] Moreover, they are not purely DAergic coming from a tumor line, because the cell line was obtained as a neuroblastoma, and this can influence genomic, metabolic or growth differentiation, moving away from the DAergic phenotype. Therefore, retinoic acid (RA) has been used to induce SH-SY5Y differentiation into neuronal phenotype, since it causes the inhibition of cell growth and increased norepinephrine production by SH-SY5Y cells [16]; furthermore, SH-SY5Y cells also express dopamine transporter (DAT) that regulates DAn homeostasis [17]. In our previous work [18] we showed that SH-SY5Y 3D spheroids express βIII-tubulin neural marker. The 1-methyl-4-phenyl-1,2,3,6-tetrahydropyridine is the prodrug of 1-methyl-4-phenylpyridinium (MPP^+^), acting on mitochondrial complexes, causing adenosine triphosphate (ATP) depletion, indirectly stimulating reactive species of oxygen (ROS) production, causing DAn apoptosis with the spill of dopamine from synaptic vesicles, resulting in the induction of apoptosis in DAn [19]. MPP^+^ in non-human primates causes the onset of an experimental Parkinsonism, which is to date the most like human idiopathic PD. These deficiencies involve cell loss in the pars compacta and lower production of serotonin, dopamine and serotonin in the striate nucleus [20]. Reduced activity of mitochondrial complexes and increased level of vulnerability to MPP^+^ have also been observed in cytoplasmic hybrids with mitochondrial DNA from PD patients [21]. Another study was conducted using an SH-SY5Y cells in vitro model for microRNA-505 studies of the function in the MPP^+^-inducing cytotoxicity. In this regard, SH-SY5Y cells have been treated with MPP^+^ to induce PD cytotoxicity [22]. Furthermore, the sensitivity of RA-treated SH-SY5Y to MPP^+^ has been tested to study dysfunctions in PD [23].

### 1.1. Pituitary Adenyate Cyclase Activating Polypeptide as Neuroprotective Peptide

Neuropeptides are signaling molecules in the CNS, and act like neurotransmitters, hormones, and neuromodulators. Like neurotransmitters, they can modulate many different processes simultaneously. Human studies revealed that the lack of neuropeptide regulation plays a pivotal role in the pathologies, such as PD [24]. Pituitary adenylate cyclase-activating polypeptide (PACAP) is a neuroprotective peptide that acts in many exogenous stimuli [25]. It is a polypeptide of 27–38 amino acids, conserved of the superfamily secretin/PACAP/glucagon, of which it belongs, for length and nucleotide sequence [26], which shows a particularly high homology with the intestinal peptide vasoactive (VIP). The neuroprotective action takes place through anti-inflammatory pathways [27,28]. PACAP stimulates the transactivation of BDNF [29,30,31]. PACAP expresses its direct effects on glial cells: astrocytes, oligodendrocytes, and microglial cells [32,33,34]. In vivo and in vitro neuronal cultures, PACAP and VIP possess powerful neuroprotective effects against trauma, ischemia, or damaging substances, such as MPTP and rotenone [35]. Based on these backgrounds, PACAP can offer a remedy against many age-related disorders, such as PD [31,32,33,34,35,36,37,38,39,40]. In fact, PACAP influences the prevention of DAn degeneration, enhancing DA synthesis [41]. PACAP exerts its action thank to their three receptors (PAC1-R, VPAC_1_, VPAC_2_) associated with adenylate cyclase via cyclic adenosine monophosphate (cAMP) and activate protein kinase A (PKA), which, in turn, can activate mitogen-activated protein kinase (MAPK) pathway [42]. These receptors are expressed in different brain regions, PAC1-R represents the receptor more expressed in the brain, compared to VPAC_1_ and VPAC_2_ [16,41,43]. PACAP/PAC1-R signal a compromise in numerous neurological disorders, such as Alzheimer’s disease, PD, and stress-related psychopathologies [44,45,46]. Maasz et al., have demonstrated the neuroprotective action of PACAP in rats treated with 6-OHDA, with complications to neuronal metabolism [47]. It has been found that PAC1-R pepducins prevented the SH-SY5Y human neuroblastoma cells, treated with MPP^+^ [40,41,42,43,44,45,46,47,48]. PACAP is a small peptide that presents low half-life in the bloodstream, representing an important concern for the treatment of neurodegenerative diseases.

### 1.2. Use of the Liposomes as Nanocarriers

Liposomes are classified as spherical nanoparticles, constituting bi-phospholipid layers in which fluid compartments are separated from aqueous environment. Liposomes are considered in compliance to their surface charge (neutral, cationic, anionic) and dimensions: small, large, or giant (>0.25 µm). Liposomes are nanocarriers able to provide many molecules, such as drugs or therapeutic agent and small organic molecules. The advantage of loading the therapeutic substances into the nanoparticles for administration to the brain are: (i) improve bioavailability, (ii) sustain release of therapeutic substance, (iii) preserve curing agents against enzymatic degradation. Furthermore, these particles can be modified on their surface by various other molecules, such as coating agents, facilitating cellular uptake of the nanoparticles themselves [49] to obtain a high concentration of the drug in the brain parenchyma. The gH-625 peptide is a cell-penetrating peptide (CCPs), a cationic and amphipathic peptide used to facilitate drug delivery and able to transport many types of molecules across the membrane bilayer in vitro and in vivo, avoiding endosomal entrapment in the membrane and lysosome degradation [50,51,52,53].

### 1.3. In Vitro Fluid-Dynamic Model of PD

Based on this background, we evaluated the neuroprotective effects of PACAP loaded into liposomes, decorated with gH625 in an in vitro fluid-dynamic model of PD. The in vitro fluid-dynamic model is represented by two different bioreactors Livebox1 and Livebox2 (LB1, LB2, IVTech, Ospedaletto, PI, Italy) in which 3D SH-SY5Y cells differentiated with retinoic acid (RA) are treated with MPP^+^. In a recent piece of work [18], we have used a LB2 bioreactor to recreate a blood–brain barrier (BBB) in vitro fluid dynamic model, placing mouse endothelial brain cells (bEnd.3) on a porous membrane in the upper chamber and 3D SH-SY5Y cells enriched to the neural portion, in the lower chamber, to demonstrate the passage of gH625-liposomePACAP-Rho through the endothelial monolayer in the lower chamber. These bioreactors are made in polydimethylsiloxane (PDMS) that ensure oxygen permeability and adequate channels for cell culture thanks to the soft lithography techniques. The membrane (20 mm), in which bEnd.3 cells are seeded, is made in polyester (PET) and present low shear stress when the bioreactor is connected to a peristaltic pump, LiveFlow (IVTech), keeping high porosity and chemical resistance [54].

Through molecular investigations, we were able to show how PACAP can act against MPP^+^ effects.

## 2. Materials and Methods

### 2.1. Peptide Synthesis

The gH625(Ac-HGLASTLTRWAHYNALIRAF-Cys) and PACAP27 (HSDGIFTDSYSRYRKQMAVKKYLAAVL-CONH2) peptides were synthesized thanks to a standard Fmoc solid-phase (GL Bio chem Ltd., Shanghai, China) [55] with an amid resin (0.6258 mmol/g of loading substitution) and good yields of about 40%. PACAP27 was also labelled with rhodamine agent (5(6)-Carboxytetramethylrhodamine N-succinimidyl ester) (Iris-Biotech GMBH, Marktredwitz, Germany) for fluorescence measurements [56,57]. A total of 20% *v*/*v* piperidine in DMF for 10 min was used to remove Fmoc protecting group and amino acids. The 2 eq Fmoc-protected amino acid, 2 eq DIC and 2 eq oxymapure was executed for the first coupling (Iris Biotech GMBH). The 2 eq amino acid, 2 eq HATU and 4 eq DIPEA was executed for the second coupling (Iris Biotech GMBH). An acid solution composed by trifluoroacetic (95%) and scavengers was performed to cleave peptides. Ice-cold diethyl ether was employed to precipitate the peptides and, subsequently, they were purified using a Phenomenex Jupiter 4µm Proteo 90 Å 250 × 21.20 mm column, with a solution composed by solvent B (linear gradient-LG) of 0.1% TFA in acetonitrile) in solvent A (0.1% TFA in water) from 20 to 80% at UV detection at 210 nm (time; 20 min). Peptides identity were established by ESI LC-MS. Peptides were cleaved and deprotected from the resin with the previous acid solution (trifluoroacetic acid and scavengers). Peptides were then purified using HPLC with preparative reverse-phase and two different solvent mixtures named solvent A with H_2_O and 0.1% trifluoroacetic acid and solvent B with CH_3_CN and 0.1% trifluoroacetic acid (LG B over 20 min with 15 mL/min of flow rate). LTQXL linear ion trap mass spectrometer (Thermo Fisher Scientific, Waltham, MA, USA) was employed to establish peptide identity. DSPEPEG2000-gH625 synthesis was obtained with a reaction of 1 eq of DSPE-PEG2000-Mal (Avant Polar Lipids, Birmingham, AL, USA), 1 eq. of pure gH625-Cys in DMF and 5 eq of triethylamine (5 eq) overnight. The reaction was monitored by RP-HPLC was used to monitor the reaction and to observe solvent evaporation. ESI LC-MS and LTQ-XL analyze the product

### 2.2. Liposome Preparations

Large unilamellar vesicles (LUVs) derived from PPC/Chol (70/30 mol/mol) were obtained in the protocol, as previously reported [18]. Briefly, chloroform was used to dissolve all lipid mixture (lipid concentration 1 mM), SPE-PEG2000-gH625 and PACAP-Rho. Nitrogen gas stream was used to remove the solvent and to carry out a lipid film, the sample was lyophilized for 24 h. LUVs were obtained as follows: suspension in Hepes 5 Mm pH 7.2 buffer, freeze–thawed 8 times and 10 times of extrusion with 0.1 μm diameter pores of polycarbonate membranes. (Northern Lipids, Burnaby, BC, Canada).

### 2.3. Liposome Characterization

A dynamic light scattering (DLS) (Malvern Zetasizer Nano ZS, Malven, UK) was employed to obtain the measure of the hydrodynamic diameters (DH) and polydispersity index (PDI) of gH625-lipoPACAP-Rho [18]. He–Ne laser 4 mW operating at 633 nm at scattering angle fixed at 173° and at 25° at pH 7.2 was used to perform all measurements in triplicate for each sample.

### 2.4. Cell Culture

Human neuroblastoma cells (SH-SY5Y) and murine endothelioma cells (bEnd.3) were cultivated individually in Dulbecco’s modified Eagle’s medium (DMEM, Sigma; St. Louis, MO, USA), adding 10% of fetal bovine serum (FBS, Sigma), 1% of L-glutamine (Sigma) and 2% of penicillin/streptomycin (Sigma) in an incubator at 37 °C, with 5% CO_2_ and controlled humidity. Cells were enzymatically detached with 0.25% trypsin/EDTA at 70% confluence.

### 2.5. Differentiation of SH-SY5Y Cells in DAn

SH-SY5Y cells were treated with RA (SH-SY5Y/RA) at a concentration of 10 µM every 24 h for six days [48]. Briefly, cells were detached and plated (1 × 10^5^ cells/cm^2^) in 100 mm plates for 24 h. The day after, they were treated with 10 μM retinoic acid (RA, Sigma) for 6 days following Piras’ protocol [58].

### 2.6. Prestoblue Assay

Before recreating a fluid-dynamic in vitro model of the BBB, we have used a Prestoblue viability assay (Invitrogen, Waltham, MA, USA) to evaluate the neuroprotective effects of PACAP on 2D SH-SY5Y/RA cells treated with different concentrations of the neurodegenerative toxin MPP^+^ in order to establish a PD model assay. Resazurin, reagent of Prestoblue assay, can evaluate the integrity of plasma membrane, DNA synthesis and enzymatic activity. Viable cells continuously convert resazurin, blueno fluorescent to resorufin, red fluorescent. SH-SY5Y/RA cells were gently detached with phosphate buffered-saline (PBS) and plated (5 × 10^3^ cells) in 96-well plates in 100 µL of complete DMEM culture medium for 24 h. Cells were then treated with MPP^+^ [13,49,59,60] at concentrations of 0.5 mM, 1 mM and 1.5 mM and PACAP in simultaneous (10^−6^ M to 10^−8^ M) to evaluate the neuroprotective effects of PACAP in cells treated with MPP^+^ (Sigma). After 24 h, 1:10 of Prestoblue reagent was added and the cells were kept for 10 min in an incubator at 37 °C and 5% of CO_2_. At the end of incubation, viable cells convert resazurin into resorufin, increasing the fluorescence and causing a change of color from blue to red. To measure absorbance expressed in optical densities (O.D.), a spectrophotometric reading was made at 570 nm using a plate reader (Synergy HTX Multi mode microplate reader). Absorbance of this compound is directly proportional to the metabolic activity and to the cell viability. For each experimental class, test was performed in triplicate. The same experiment was conducted for gH625-lipoPACAP-Rho to analyze the differences between PACAP action and PACAP action with liposome functionalized with gH peptide. Briefly, after establishing the PD model on SH-SY5Y/RA cells, gH625-lipoPACAP-Rho was added at a range concentration of PACAP-Rho from 10^−6^ M to 10^−8^ M, simultaneously with MPP^+^ action. For each experimental class, Prestoblue test was performed in triplicate.

### 2.7. The 3D SH-SY5Y/RA in Dynamic Culture

After evaluating the neuroprotective effects of PACAP on 2D SH-SY5Y/RA, we produced 3D SH-SY5Y/RA cells by hanging drop method (2 × 10^4^ cells per aggregate) [18,19,20,21,22,23,24,25,26,27,28,29,30,31,32,33,34,35,36,37,38,39,40,41,42,43,44,45,46,47,48,49,50,51,52,53,54,55,56,57,58,59,60,61]. After 48 h from the aggregate’s formation, they were transferred in an LB1 bioreactor. LB1 is composed by a polydimethylsiloxane (PDMS) chamber with an inlet and outlet tube connected with its mixing chamber to a peristaltic pump fluid circuit (Liveflow) [54] at a flow to 100 µL/min. Fluid allows a continuous recycle of nutrient in which spheroids are in suspension without attaching in the lower part of the chamber.

### 2.8. The 3D SH-SY5Y Immunofluorescence Assay

To assess the presence of PACAP receptors (PAC1-R, VPAC_1_, VPAC_2_) in 3D SH-SY5Y/RA cells in dynamic culture before evaluating its neuroprotective effects after 24 h of dynamic flow (LB1—100 µL/min), we conducted different indirect immunofluorescence assays. Spheroids were fixed with cold methanol for 15 min. Blocking site was performed with 3% BSA (bovine serum albumin, Invitrogen) in 0.1% Triton-PBS (Sigma) for 30 min. Primary antibody chosen were PAC1-R, VPAC_1_, VPAC_2_ (Abcam, Cambridge, UK) for 1:45 min in 1%BSA/PBS. Secondary antibody, AlexaFluor 488 (1:500 in 1%BSA/PBS, Invitrogen) was performed for 1 h. Cell nuclei were labeled with DAPI (1:1000 in PBS, Invitrogen) for 5 min. Images were acquired with the JuLi ™ Stage_RealTime Cell History Recorder microscope with 10× objective, using two different channels: GFP and DAPI. For each experimental condition, three immunofluorescence assays were repeated, and different fields were randomly selected for data analysis. The captured images were corrected for brightness and contrast using Fiji software.

### 2.9. Annexin V-FITC/Propidium Iodide Assay

Annexin V-FITC/Propidium Iodide (PI) (Biotool, Kirchberg, Switzerland) assay was carried out in 3D SH-SY5Y/RA/MPP^+^ (1.5 mM) for 24 h in fluid dynamic condition (100 µL/min) to discriminate the necrotic and apoptotic cells after MPP^+^ treatment. The MPP^+^ concentration was chosen, being the highest to create damage according to previously Prestoblue assay. Control was carried out with another LB1 containing 3D SH-SY5Y/RA not treated with MPP^+^ in the same fluid condition. Annexin V (FITC) highlights the apoptotic cells. PI (red dye) highlights necrotic cells. Spheroids are treated for 15 min with both probes (Annexin V-FITC/PI) dissolved in 1X binding buffer. Nuclei were labeled with DAPI for 5 min. Images were acquired with the JuLi ™ Stage RealTime Cell History Recorder microscope with 10× objective, using three different channels: DAPI, GFP and RFP. For each experimental condition, three different assays were repeated, and different fields were randomly selected for data analysis. The images were corrected for brightness and contrast using Fiji software.

### 2.10. Spectrofluorimetry Assay

The 3D SH-SY5Y/RA/MPP^+^ and 3D SH-SY5Y/RA were adapted to flow condition in two LB1s for 24 h. Once adapted, they were transferred in the lower chamber of three LB2s. LB2 is a double chamber (upper and lower) bioreactor with double independent inlet and outlet tube used to mimic physiological barrier [16,17,18,19,20,21,22,23,24,25,26,27,28,29,30,31,32,33,34,35,36,37,38,39,40,41,42,43,44,45,46,47,48,49,50,51,52]. The double chambers are separated by a porous membrane (0.45 µm, ipPORE, Belgium). The porous membrane ensures the passage of nutrients without spill cells, allowing good adhesion [54] (Figure 1a,b). On the porous membrane, bEnd.3 cells were seeded for 7 days to mimic a reliable and stable BBB [18]. Three LB2s were organized, as reported below:The bEnd.3 cells in the upper chamber, 3D SH-SY5Y/RA/MPP^+^ in the lower chamber.The bEnd.3 cells in the upper chamber, 3D SH-SY5Y/RA in the lower chamber.The bEnd.3 cells in the upper chamber, 3D SH-SY5Y/RA/MPP^+^ in the lower chamber.

In the first two LB2s, gH625-lipoPACAP-Rho was injected in the inlet tube of the upper chamber at a concentration of 10^−8^ M. The third LB2 represents a control check to observe the behavior of the cells MPP^+^ treated without gH625-lipoPACAP-Rho. Experiments were carried out in 24 h at 100 µL/min.

### 2.11. Protein Extraction and Western Blot

We have previously established a reliable and stable monolayer endothelial barrier using bEnd.3 cells seeded on the porous membrane of LB2 connected to a Liveflow pump circuit [18]. We have performed permeability test (Lucifer yellow assay) to determine the integrity of the barrier after 7 days of continuous flow rating. The porous membrane results in low cytotoxicity on bEnd.3 cells. Moreover, we have previously demonstrated the presence of anti ZO-1, anti N-cadherin and anti-β catenin, with an immunofluorescence assay for bEnd.3 cells on the porous membrane in order to evaluate the formation of the barrier and its integrity [18]. To evaluate the semiquantitative presence of the protein cells junction’s, in the present work, we have performed the extraction of proteins from the bEnd.3 cells placing on the porous membrane and subsequently the Western Blot. LB2 containing bEnd.3 cells on the porous membrane were placed on ice for 10 min and washed twice with cold PBS. After adding 1 mM PBS-EDTA (Sigma), cells were mechanically lysed. Cell lysates were obtained using RIPA lysis buffer and a cocktail of phosphatase inhibitors (Santa Cruz Biotechnology, Dallas, TX, USA). Protein concentrations were determined by BCA assay (Pierce Biotechnology, Rockford, IL, USA). Proteins were separated by 10% SDS-PAGE and transferred onto the PVDF membrane using a trans-blot (Bio-Rad, Hercules, CA, USA), at a constant amperage of 220 mA for 45 min. After proteins were transferred, they were incubated in a 7% Bovine Serum Albumin (BSA, Sigma) solution in TBS-T (TBS 0.05% Tween-20, Sigma) for 1 h at room temperature and then overnight at 4 °C with the primary antibodies ZO-1 (1:300) (Rabbit-Abcam), N-Cadherin (1:500) and β-catenin (1:500) (Rabbit-Santa Cruz Biotechnology, Dallas, TX, USA) diluted in 5% BSA solution in TBS-T (TBS 0.05% Tween-20). Membranes were washed 3 times with TBS-T and incubated for 1 h with the secondary anti-rabbit IgG antibody (Santa Cruz Biotechnology) conjugated with horseradish peroxidase diluted 1: 4000 in 3% BSA solution in TBS-T (TBS 0.05% Tween-20). Subsequently, the PVDF membranes were washed, and the bands related to cells junctions’ protein were detected with chemiluminescence (Santa Cruz Biotechnology) using the C-DiGit scanner (LI-COR) and the Image Studio software to determine their optical density (OD). Optical density was normalized, with respect to β-actin (1:500) in 5% BSA solution in TBS-T (TBS 0.05% Tween-20). Data were obtained from three independent Western Blots.

### 2.12. Quantitative Measurement of Reactive Oxygen Species (ROS)

Cells are continuously exposed to reactive oxygen species (ROS) as a product of respiratory metabolism and the amount in ROS production may be due to an increase in mitochondrial activity. Cells, subjected to external stress (thermal shock, high osmolarity), can produce ROS becoming metabolically less efficient. In conditions of excessive stress, the quantity of ROS produced exceeds the antioxidant capacity of the cell, leading to oxidative damage [62,63]. Oxidative stress can be quantified using dichloro dihydro fluoresceine diacetate (DCFH2-DA) [64,65]. It is a fluorescent probe that crosses cell membranes and is trapped by intracellular esterase in the cytoplasm. The presence of oxidizing species oxidizes dichlorofluorescein diacetate to 2,7dichlorofluorescein (DCF), a highly fluorescent molecule. In this test, spheroids (3D SH-SY5Y/RA/MPP^+^ and 3D SH-SY5Y/RA with and without gH625-lipoPACAP-Rho) were incubated with 50 μM DCFH2-DA (Thermo Fisher Scientific) for 60 min (37 °C, 5% CO_2_) [62]. Positive control was performed by adding 2 mM of H_2_O_2_ to the spheroids for 90 min [63]. To measure fluorescence, spheroids were first lysed with 1 mM PBS/EDTA (Invitrogen) and then a spectrophotometric reading was performed at the excitation wavelength (Ex) of 485 nm and at the emission wavelength (Em) of 528 nm using a plate reader (Synergy HTX multimode microplate reader). Three assays were performed and for each experimental class the test was performed in triplicate.

### 2.13. Prestoblue Assay

Prestoblue assay was performed on the spheroids present in the three LB2s descripted in 2.9. This assay was useful to determine cell viability on spheroids treated or not with MPP^+^ after the passage of gH625-liposomePACAP-Rho. Briefly, after 24 h, spheroids were transferred in a 96-well plate and Prestoblue cell reagent was added for 4 h [66,67]. Absorbance was measured at 570 nm using a plate reader (Synergy HTX multimode microplate reader). Three assays were performed and for each experimental class, the test was performed in triplicate.

### 2.14. Statistical Analyses

All experiments were performed in triplicate, and the data were expressed as means ± SEM. Statistical analysis was performed through the analysis of variance (ANOVA) and the Dunnet’s post-test. The two-tailed Mann–Whitney test was performed to evaluate treated groups, compared with the appropriate control group.

## 3. Results

### 3.1. Liposome Characterization

The gH625-lipoPACAP-Rho was characterized with a dynamic light scattering (DLS), as reported in our previous work [18]. Each sample was performed independently in triplicate. The gH625-lipoPACAP-Rho shows a mean diameter of 193.9 ± 8.050 nm. All the samples present a polydispersity index (PDI) < 0.3 (good size distribution).

### 3.2. Prestoblue Assay

The neuroprotective effect of PACAP was evaluated using the Prestoblue assay on 2D SH-SY5Y neuroblastoma cells, differentiated into DAn (SH-SY5Y/RA) and treated with the neurodegenerative agent, MPP^+^ (0.5 mM, 1 mM, and 1.5 mM). The concentrations of PACAP that were used included a range of values from 10^−6^ M to 10^-8^ M. The treatment was carried out with PACAP and MPP^+^ simultaneously administered. Control cells were represented by SH-SY5Y/RA (RA). After 24 h, results show a significant increase in cell viability in the presence of PACAP (Figure 2).

The increase in cell viability can also be observed when DAn/MPP^+^ are treated with gH625-lipoPACAP-Rho ranging from 10^−6^ M to 10^−8^ M of PACAP, although the increase is not significant at 0.5 mM for 10^−7^ and 10^−8^ M and at 1.5 mM MPP^+^ for 10^−7^ M PACAP, respectively (Figure 3).

Results obtained show higher viability, compared to control, at all concentrations tested. This suggests that gH625-liposome does not hinder PACAP activity, rather it contributes to viability at the analyzed concentrations (10^−6^ M to 10^−8^ M).

### 3.3. The 3D SH-SY5Y Immunofluorescence Assay

Different indirect immunofluorescence assays were performed to evaluate the presence of PACAP receptors (PAC1-R, VPAC_1_, VPAC_2_) in 3D SH-SY5Y after 24 h of flow condition (100 μL/min). After 1 h and 45 min of antibody incubation, 3D SH-SY5Y cells showed an evident fluorescence signal for and PAC1-R (Figure 4a), while the signal for VPAC_1_ (Figure 4b) and VPAC_2_ (Figure 4c) was less evident, suggesting, as shown in the literature, a major presence of PAC1-R in this cell line [68,69,70,71]. Nuclei were stained with DAPI.

### 3.4. Annexin V-FITC/Propidium Iodide Assay

The 3D SH-SY5Y/RA/MPP^+^ and 3D SH-SY5Y/RA after 24 h in fluid dynamic condition (100 µL/min) in LB1 were placed in the JuLi ™ Stage Real-Time Cell History Recorder (Figure 5, Figure 6 and Figure 7).

Annexin V-FITC/PI assay to 3D SH-SY5Y/MPP^+^ shows a majority presence of necrotic cells (Necro) after 24 h (Figure 6).

The 3D SH-SY5Y, instead, shows a majority presence of apoptotic cells (Apo) (Figure 8). This different behavior is probably due to MPP^+^ action that acts on death of neurons, causing cells necrosis [21,22,23].

### 3.5. Spectrofluorimetry Assay

Spectrofluorimetry assay was performed to evaluate the capability of gH625-liposome to carry PACAP across endothelial brain cells monolayer (bEnd.3) in two LB2s; one of them contained in the lower chamber 3D SH-SY5Y/RA without MPP^+^ (Figure 9).

In another one, LB2 was in the upper chamber bEnd.3 cells, seeded on the porous membrane and in the lower chamber 3D SH-SY5Y/RA, treated with 1.5 mM MPP^+^ (Figure 10).

After 24 h with the gH625-lipoPACAP-Rho injection, in these LB2, the amount of PACAP-Rho is higher in the lower chamber (LC) than the upper chamber (UC), indicating that PACAP crossed endothelial brain cells monolayer in LB2 and reached the lower chamber.

### 3.6. Western Blot

After carrying out the protein extraction, the expression of the cells junction’s protein was evaluated through Western Blot. Samples collected of the bEnd.3 cells on the porous membrane of Livebox2 were subjected to SDS–polyacrylamide gel electrophoresis and Western blotting. Semiquantitative densitometric analysis revealed the presence of bands related to ZO-1 (corresponding to its molecular weight: 195 kDa), N-Cadherin (corresponding to its molecular weight: 132 kDa) and β-catenin (corresponding to its molecular weight: 85–95 kDa) (Figure 11).

The stripping of the membrane blot and its re-probing with β-actin (molecular weight: 45 kDa) polyclonal antibody showed equal loading of proteins in all lanes (Figure 12). The BBB acts as selectively structure that protects brain from external injury, thanks to the presence of junctions’ protein among brain endothelial cells, ensuring the low paracellular permeability. For this reason, their expression can change in response to many diseases [68,69]. In particular, low levels of ZO-1 protein in the 6-OHDA and MPTP models have been studied [72,73,74]. For this reason, we have evaluated different presence of three junctions’ protein: anti-ZO1, N-cadherin, and anti-β catenin as their maximum expression indicates a stable barrier. Since the tight junction proteins are more expressed in the permeability of the barrier, the high presence of ZO-1, compared to the adherent junctions, indicates a stable barrier that allows us to act by performing the liposomal passage.

### 3.7. Quantitative Measurement of Reactive Oxygen Species (ROS)

To carry out a quantitative measurement of reactive oxygen species, dichlorodihydrofluoresceindiactetate (DCFH2-DA) was used. After incubation with 50 μM DCFH2-DA for 60 min, spheroids (3D SH-SY5Y/RA/MPP^+^ and 3D SH-SY5Y/RA with and without gH625-lipoPACAP-Rho) were lysed by PBS/EDTA to perform the fluorescence measurement. Positive control was performed by adding 2 mM of H_2_O_2_ to the spheroids for 90 min [63]. The graph (Figure 13) shows a decrease in fluorescence (less ROS concentration) in spheroids treated with gH625-lipoPACAP-Rho (PACAP) and both with gH625-lipoPACAP-Rho and MPP^+^ (PACAP MPP^+^), compared to the spheroids treated only with MPP^+^ (MPP^+^) and to positive control (PC). The values of the untreated cells (3D SH-SY5Y/RA) were calculated as baseline by subtracting them from the indicated values.

### 3.8. The 3D Prestoblue Assay

Prestoblue assay was performed to evaluate cell viability on 3D SH-SY5Y/RA/MPP^+^ and 3D SH-SY5Y/RA/MPP^+^ treated with gH625-lipoPACAP-Rho and 3D SH-SY5Y/RA treated with gH625-lipoPACAP-Rho. Results obtained show a higher viability in both LB2s with gH625-lipoPACAP-Rho, compared to spheroids treated only with MPP^+^. Moreover, it is of note that viability is higher when cells are both treated with gH625-lipoPACAP-Rho and MPP^+^, compared to only gH625-lipoPACAP-Rho (Figure 14).

## 4. Discussion

PD is one of the most important neurodegenerative disease-causing movement disorders due to death of DAn in the substantia nigra [75]. Despite the progress in the scientific area, to date there is no real cure for these pathologies, except for the use of symptomatic drugs [76]. Therefore, it is mandatory to develop more therapeutic agents to counter long-term effects. Moreover, the cause of PD is unknown, despite many factors involved in its development, including genetic and environmental factors. The human neuroblastoma cell line, SH-SY5Y, has been widely used as in vitro model of PD studies. They can acquire neuron-like phenotypes when treated with acid retinoic (RA) [77]. RA has a pivotal role to maintain growth and differentiation of cells from proliferating precursor cells to post mitotic differentiated cells MPP^+^ is a neurotoxin that can cause the onset of an experimental parkinsonism in SH-SY5Y cells [12,13], involving a reduction in dopamine, noradrenaline and serotonin. PACAP acts with its specific receptors (PAC1-R, VPAC_1_, VPAC_2_), which are shared with VIP [38]. PACAP can act as neurotransmitter, neuromodulator, or neuroprotective agent against more toxic substances, such as MPTP, or rotenone in both in vivo and in vitro culture [75]. PACAP can offer a therapeutic approach in the treatment of PD [27,33,34,37] but it also presents low bioavailability in the bloodstream. Nanoparticles, such as liposome, represent good candidates to carry therapeutic molecules in brain parenchyma, thanks to their low availability, small size (6–300 nm) and low cost. They can also be modified on their surface by other penetrating peptides to facilitate drug uptake in the brain [47]. The gH625 is a perturbing membrane domain of 19 amino acid residues (625aa to 644aa) derived from the H (gH) glycoprotein of the *Herpes simplex* 1 virus [48,49]. The aim of this work is to study the neuroprotective effects of PACAP loaded with gH625-liposome in an in vitro fluid-dynamic model of PD. We have realized a BBB in fluid-dynamic conditions, with the help of a bioreactor named LB2. This bioreactor is equipped with two different chambers: upper and lower ones, each of which possess two independent inlets and outlets. In the upper chamber, on the porous membrane, we have seeded a monolayer by endothelial brain cells (bEnd.3) and in the lower chamber we have seeded 3D SH-SY5Y/RA both treated and not with a neurodegenerative toxin, MPP^+^, in order to evaluate the passage and the neuroprotective effect of PACAP, delivered by gH625-liposome. Previously [18], we have demonstrated the gH625-lipoPACAP-Rho passage in a BBB fluid-dynamic with bEnd.3 cells in the upper chamber and 3D SH-SY5Y/RA in the lower ones of the LB2. The goal of this work is to evaluate the passage of liposome functionalized when 3D SH-SY5Y are treated with MPP^+^ and the potential PACAP neuroprotective effects. In our fluid-dynamic model, the flow conditions do not alter endothelial cell layer health with a stable expression of junction proteins [18]. Moreover, 3D SH-SY5Y spheroids express PACAP receptors, particularly PAC1-R, as already reported in the literature [69,70,71]. These data allowed us to perform delivery experiments through the endothelial layer down to PACAP-sensitive SH-SY5Y cells. Our data on cell viability Prestoblue assay conducted on 2D SH-SY5Y/RA cells (treated with MPP^+^) show that PACAP can act as neuroprotective agent against different MPP^+^ concentrations (0.5 mM, 1 mM, 1.5 mM). As widely reported in the literature [41,42,43,44,45,46,47,48,49,50,51,52,53,54,55,56,57,58,59,60,61,62,63,64,65,66,67,68,69,70,71,72,73,74,75,76], PACAP can act as reducing cell death in different neuroblastoma cell lines, such as SH-SY5Y, when treated with different neurotoxins preserving membrane cells and mitochondrial activity [78]. At the tested MPP^+^ concentrations, PACAP acts as a neuroprotector throughout the physiological concentration range used (10^−6^ M to 10^−8^ M); this is in good agreement with literature [42]. The most important concern for treating neurological disorders, such as PD, is to enhance its bioavailability in the bloodstream and then in the brain. For this reason, we used PACAP loaded within gH625 functionalized liposome (LPACAP) in order to evaluate if the neuroprotective effect of PACAP was influenced by the nanodelivery tool. LPACAP always produces an increase in cell viability, compared to MPP^+^, thus, LPACAP is biologically active even if not all concentrations tested were statistically significant (i.e., 10^−7^ M). Although the partial discrepancy remains to be elucidated, results strongly suggest, along with the previously demonstrated effectiveness of gH625 liposomes to deliver PACAP to the brain [53], that LPACAP has a robust neuroprotective action on neuronal cells and that our delivery system does not substantially represent an obstacle to the PACAP activity, also guaranteeing PACAP bioavailability [18]. To better understand PACAP neuroprotective action, we have used 3D SH-SY5Y with enriched neural portion (3D SH-SY5Y/RA) and treated with MPP^+^. Our molecular analysis showed an increasing number of necrotic cells in 3D SH-SY5Y/RA/MPP^+^. In a different scenario, when 3D SH-SY5Y/RA/MPP^+^ was treated with gH625-lipoPACAP-Rho, we noted an increased number of apoptotic cells, suggesting that PACAP action does not lead to a neuronal death but causes the cells to remain in apoptosis. The 3D SH-SY5Y/RA cells also show sensitivity to PAC1-R, maybe as PACAP can act to restore cells using PAC1-R pathway. Moreover, when 3D SH-SY5Y/RA/MPP^+^ are seeded under brain endothelial cells monolayer to mimic BBB, we showed that high amounts of PACAP can be detected in the neuronal compartment, suggesting that our nanodelivery system efficiently delivers PACAP in vitro, as previously reported [18]. In our previous work, gH625-lipoPACAP-Rho can cross endothelial brain monolayer without being retained by bEnd.3 cells [18]. We previously showed that gH625-lipoPACAP-Rho in vitro BBB crossing took place after 30 min of injection [18]. To verify the neuroprotective action, we increased experimental time to 24 h. It is of note that we recorded higher amounts of PACAP in the lower chambers throughout the experiments, indicating that the in vitro system does not reach the equilibrium state (i.e., the same amounts of PACAP in both chambers), although we did not verify the hypothesis, this could be due to active transportation mechanisms acting on endothelial cells. This is in accordance with the increased ability of gH625 to act as an efficient delivery cargo with low cytotoxicity, as previously demonstrated [18,50,51,52,53]. This finding is further important since in vivo PACAP can be easily transported, via efflux pump, out of the brain. Hence, after gH625-lipoPACAP-Rho administration, the quick transportation through the bEnd.3 cells monolayer ensures appropriate amounts of PACAP being delivered to the neuronal compartment. The role of PACAP as an antioxidant, acting also on ROS production, has been widely reported [79,80,81,82]. Decrease in spheroids ROS production, treated with gH625-lipoPACAP-Rho, demonstrates that the enhanced delivery of PACAP is useful to protect neuronal 3D SH-SY5Y cells from MPP^+^-induced damage. This is also supported by the evidence that neuronal cell viability is recovered to physiological values when cells, damaged by MPP^+^, are treated with gH625-lipoPACAP-Rho.

## 5. Conclusions

We evaluated the neuroprotective effects of PACAP, delivered by a functionalized liposome-based enhanced drug delivery tool, using an in vitro fluid dynamic model of BBB/PD. Our data demonstrated both the efficacy of the delivery system and the neuroprotective effects of PACAP, when injected through the compartment mimicking the blood flow, on compartment mimicking the neuronal parenchyma and preserving the neurons health. In conclusion, the evidence supports the effectiveness of our functionalized nanosystem to improve the neuronal bioavailability of the neuroprotective agent, such as PACAP, when administered systemically. This is promising for the development of a clinical solution based on the PACAP neuroprotection in the treatment of PD.

## Figures and Tables

**Figure 1 biomedicines-10-02644-f001:**
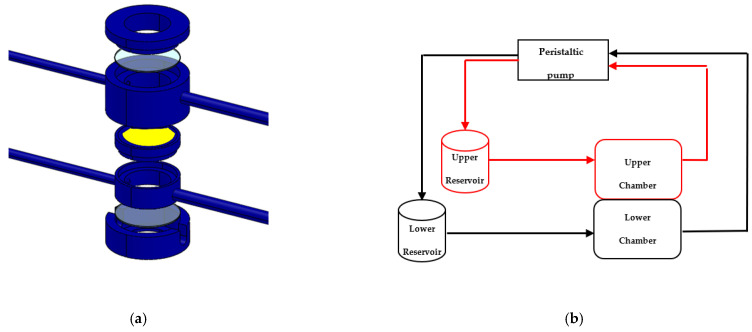
The 3D schematic exploded view of Livebox2 dynamic bioreactor composed by an upper chamber and a lower chamber separated by a porous membrane (yellow) with an appropriate rounded microscopy coverslip (**a**). Chambers of the bioreactor are connected to their respective reservoirs and to a peristaltic pump (**b**).

**Figure 2 biomedicines-10-02644-f002:**
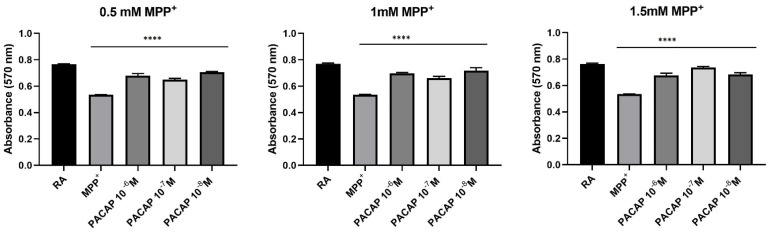
Spectrophotometric analysis on the effects of PACAP on cell viability following MPP^+^ treatment on dopaminergic neurons (SH-SY5Y/RA). The results show exposure after 24 h. The graph shows the means ± SEM of three experiments. Statistical analysis was performed through the analysis of variance (ANOVA) and the Dunnet’s post-test. Values were considered significant, compared to the control MPP^+^. **** *p* < 0.0001.

**Figure 3 biomedicines-10-02644-f003:**
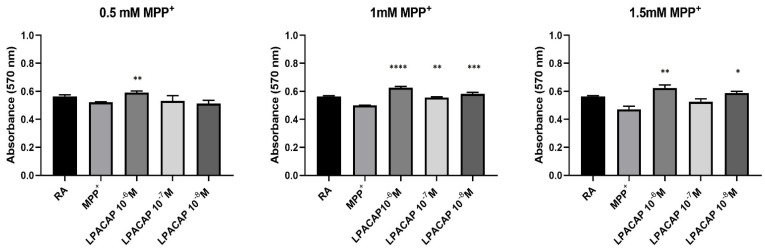
Spectrophotometric analysis on the effects of gH-lipoPACAP-Rho (LPACAP) on cell viability following MPP^+^ treatment on dopaminergic neurons (SH-SY5Y/RA). The results show exposure after 24 h. The graph shows the means ± SEM of three experiments. Statistical analysis was performed through the analysis of variance (ANOVA) and the Dunnet’s post-test. Values were considered significant, compared to the control MPP^+^. * *p* < 0.05, ** *p* < 0.01, *** *p* < 0.001, **** *p* < 0.0001.

**Figure 4 biomedicines-10-02644-f004:**
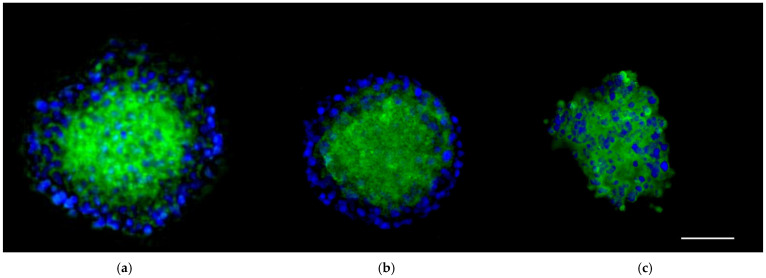
Representative image of spheroids obtained by culturing human neuroblastoma 3D SH-SY5Y/RA cells in the LB1 after 24 h of fluid condition (100 μL/min). Spheroids were immunoreacted with PAC1-R primary antibody (GFP, (**a**)), VPAC_1_ primary antibody (GFP, (**b**)), VPAC_2_ primary antibody (GFP, (**c**)). Nuclei, DAPI. Images were acquired as Z stack with the JuLI Stage fluorescence recorder and maximum intensity projection was shown. Scale bar: 250 µm.

**Figure 5 biomedicines-10-02644-f005:**
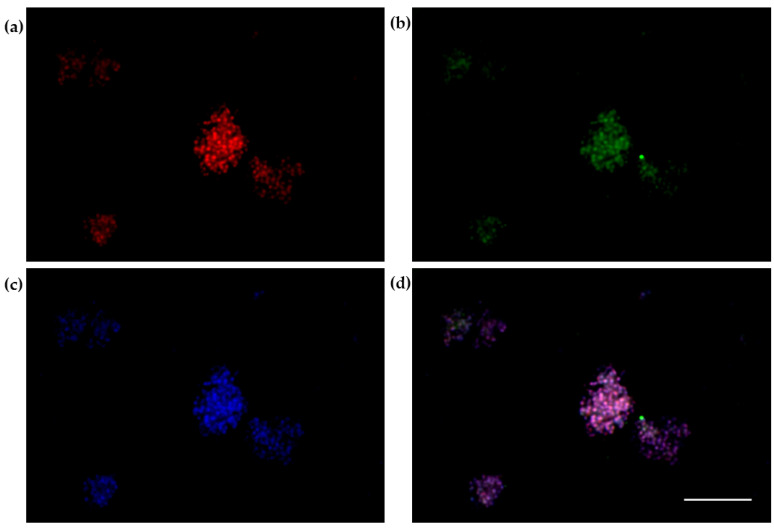
Annexin/PI labelling of neural spheroids treated with 1.5 mM MPP^+^: apoptotic cells ((**a**)-red); necrotic cells ((**b**)-green); DAPI ((**c**)-blue); Merge (**d**). Scale bar: 100 µm.

**Figure 6 biomedicines-10-02644-f006:**
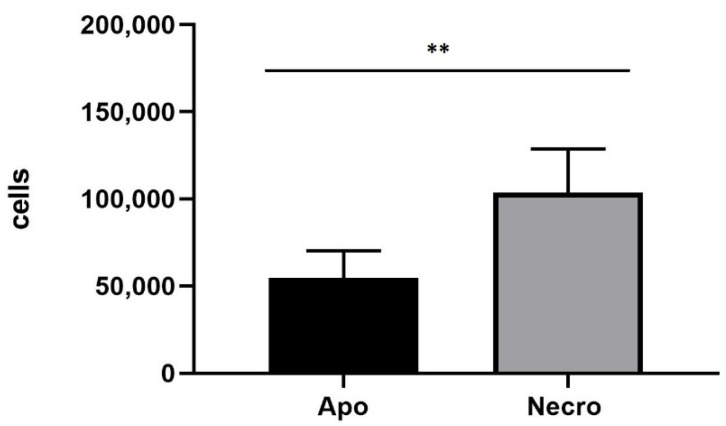
Annexin/PI on neural spheroids (3D SHSY5Y/RA) treated with 1.5 mM MPP^+^. The number of necrotic cells is higher, compared to apoptotic cells. The graph shows the means ± SEM of three experiments. Statistical analysis was performed through the analysis of variance (ANOVA) and the Dunnet’s post-test. The differences were statistically significant between apoptotic and necrotic cells. ** *p* < 0.01.

**Figure 7 biomedicines-10-02644-f007:**
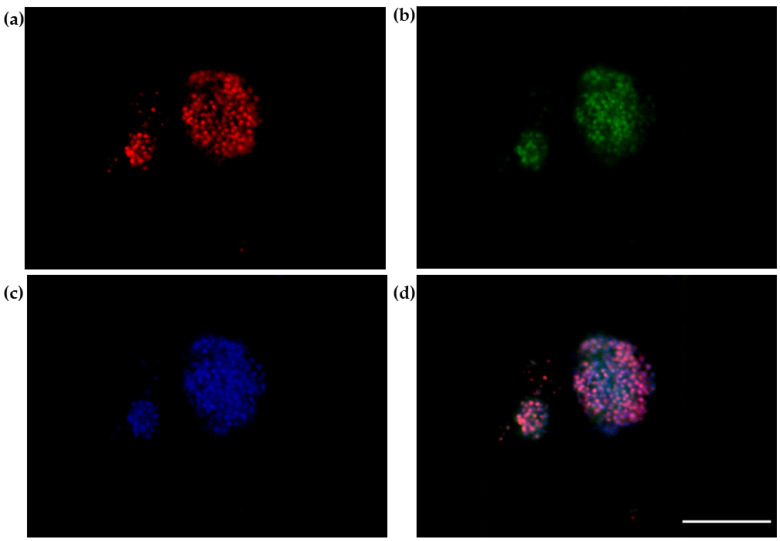
Annexin/PI labelling of neural spheroids without 1.5 mM MPP^+^: apoptotic cells ((**a**)-red); necrotic cells ((**b**)-green); DAPI ((**c**)-blue); Merge (**d**). Scale bar: 100 µm.

**Figure 8 biomedicines-10-02644-f008:**
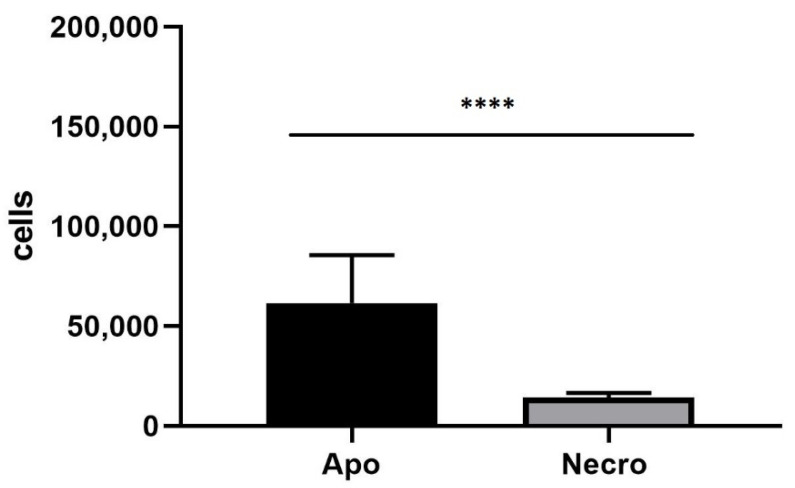
Annexin/PI on neural spheroids without 1.5mM MPP^+^. The number of apoptotic cells is higher compared to necrotic cells. The graph shows the means ± SEM of three experiments. Statistical analysis was performed through the analysis of variance (ANOVA) and the Dunnet’s post-test. The differences were considered significantly between apoptotic and necrotic cells. **** *p* < 0.01.

**Figure 9 biomedicines-10-02644-f009:**
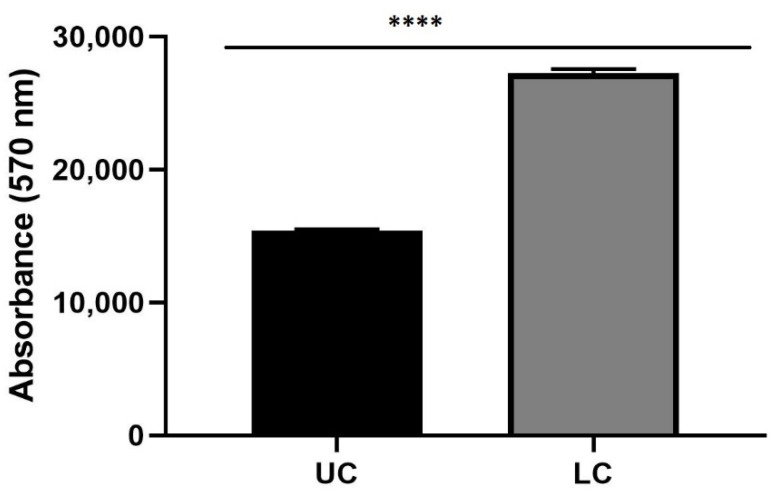
Spectrofluorometric analysis of rhodaminated PACAP (gH625-lipoPACAP-Rho) delivery across a BBB dynamic in vitro model (LB2) with bEnd.3 in the upper chamber and 3D SH-SY5Y/RA in the lower chamber. Functionalized liposome was loaded with PACAP-Rho and injected in the upper flow. The passage beyond the endothelial cell layer were then evaluated by sampling downstream the upper chamber (UC) and the lower chamber (LC), respectively. The graph shows the means ± SEM of three experiments. Statistical analysis was performed through the analysis of Mann–Whitney test. The differences were considered significant between gH625-lipoPACAP-Rho UC and gH625-lipoPACAP-Rho LC. **** *p* < 0.001.

**Figure 10 biomedicines-10-02644-f010:**
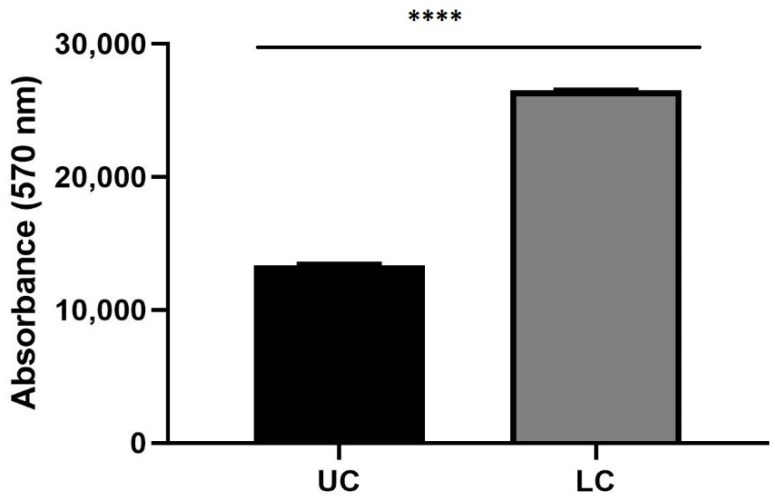
Spectrofluorometric analysis of rhodaminated PACAP (gH625-lipoPACAP-Rho) delivery across BBB dynamic in vitro model (LB2) with bEnd.3 in the upper chamber and 3D SH-SY5Y/RA/MPP^+^ in the lower chamber. Functionalized liposome was loaded with PACAP-Rho and injected in the upper flow. The passage beyond the endothelial cell layer was then evaluated by sampling downstream the upper chamber (UC) and the lower chamber (LC), respectively. The graph shows the means ± SEM of three experiments. Statistical analysis was performed through the analysis of Mann–Whitney test. The differences were considered significant between gH625-lipoPACAP-Rho UC and gH625-lipoPACAP-Rho LC. **** *p* < 0.001.

**Figure 11 biomedicines-10-02644-f011:**
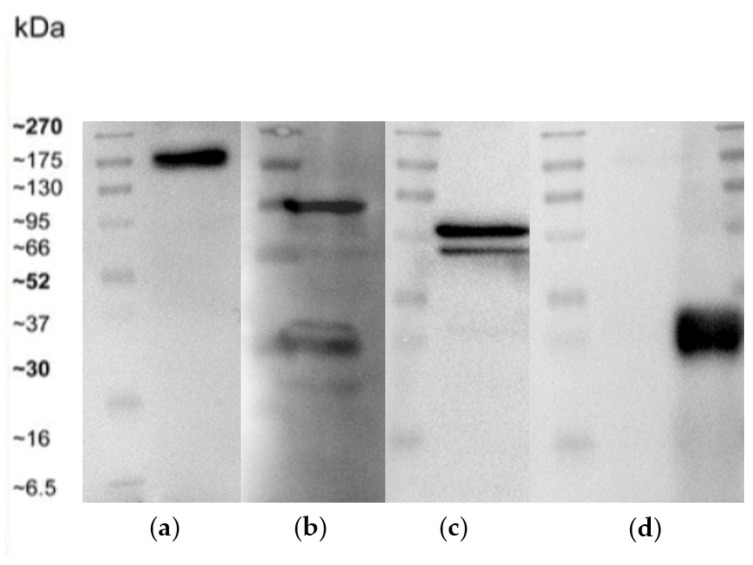
Expression of cell junctions’ proteins in bEnd.3 cells cultured on the porous membrane of Livebox2 after one week of flow conditions (100 µL/min). Western blotting analysis: lane 1 (**a**), ZO-1 tight junctions’ expression; lane 2 (**b**), N-Cadherin adherens junctions’ expression; lane 3 (**c**) β-catenin adherens junctions’ expression. The blot was stripped and re-probed with an anti-β-actin polyclonal antibody to ensure equal loading of proteins in the remaining lanes (**d**). Molecular mass markers are indicated on the left of the Western blotting panels.

**Figure 12 biomedicines-10-02644-f012:**
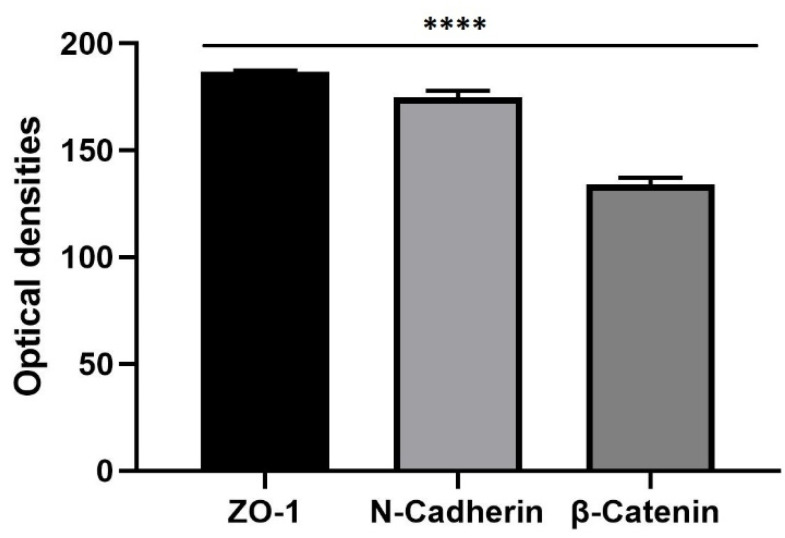
Semiquantitative analysis of cell junctions’ proteins in bEnd.3 cells cultured on the porous membrane of Livebox2 after one week of flow conditions (100 µL/min). Cell lysates were normalized, with respect to anti-β-actin polyclonal antibody. ZO-1, tight junctions’ protein results expressed more than N-Cadherin and β-Catenin, adherens junctions proteins. The graph shows the means ± SEM of three different Western blotting. Statistical analysis was performed through the analysis of Mann–Whitney test. The differences were considered significant among junctions’ proteins. **** *p* < 0.001.

**Figure 13 biomedicines-10-02644-f013:**
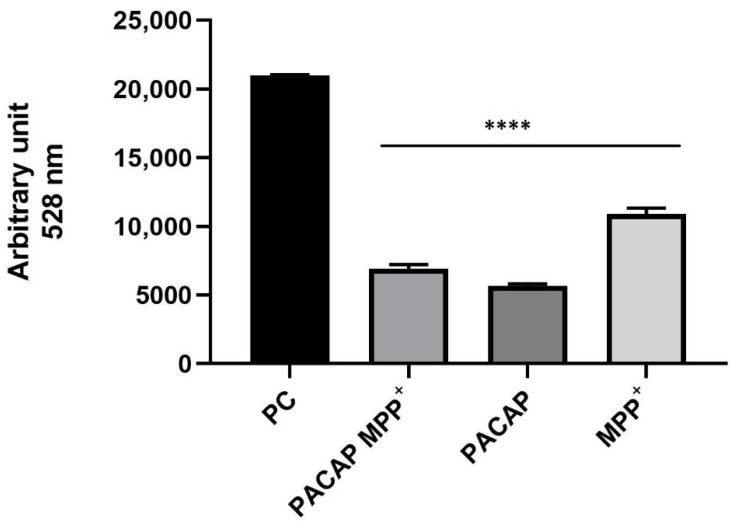
DCFH2-DA assay on 3D SH-SY5Y/RA treated with both gH625-lipoPACAP-Rho and MPP^+^ (PACAP MPP^+^), 3D SH-SY5Y/RA treated with gH625-lipoPACAP-Rho (PACAP) and 3D SH-SY5Y/RA treated with MPP^+^ (MPP+). Spheroids treated with gH625-lipoPACAP-Rho show a decrease in ROS concentration, mainly compared to MPP^+^. The graph shows the means ± SEM of three experiments. Statistical analysis was performed through the analysis of variance (ANOVA) and the Dunnet’s post-test. The differences were considered significant, compared to the MPP^+^. **** *p* < 0.01.

**Figure 14 biomedicines-10-02644-f014:**
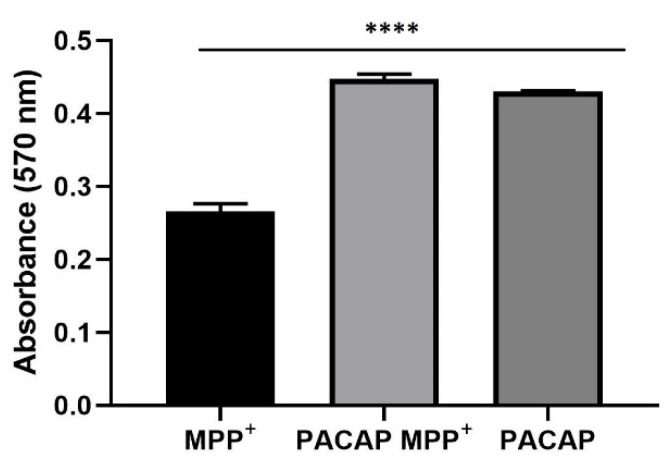
Prestoblue assay on 3D SH-SY5Y/RA/MPP^+^ (MPP^+^) and 3D SH-SY5Y/RA/MPP^+^ treated with gH625-lipoPACAP-Rho (PACAP MPP^+^) and 3D SH-SY5Y/RA treated only with gH625-lipoPACAP-Rho (PACAP). The 3D SH-SY5Y/RA spheroids show higher viability when treated with gH625-lipoPACAP-Rho (PACAP MPP^+^ and PACAP), compared to 3D SH-SY5Y/RA/MPP^+^ (MPP^+^). The graph shows the means ± SEM of three experiments. Statistical analysis was performed through the analysis of variance (ANOVA) and the Dunnet’s post-test. The differences were considered significant, compared to spheroids (3D SH-SY5Y/RA) treated with MPP^+^ (MPP^+^). **** *p* < 0.01.

## Data Availability

Not applicable.

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
