# Peer review of "Neuroprotective Effects of gH625-lipoPACAP in an In Vitro Fluid Dynamic Model of Parkinson’s Disease"

_biomedicines, 2022, doi:10.3390/biomedicines10102644_

Round 1

Reviewer 1 Report

The work by Barra et al is certainly important and generally carefully done. However, before being published, there are issued that I think might be good to solve:

1. There are minor scientific English issues that the authors should revise (e.g. Western blot should be written with capital letter always, etc.).

2. Figure legeneds (multiple) should be independent and clear for the reader. Where is RA coming from?

"Spectrophotometric analysis on the effects of PACAP on cell viability following MPP+ 308
treatment on dopaminergic neurons (RA)"?

3. If one carefully looks in Figure 5, there are many same cells marked both for apoptosis and necrosis. I do not see that clearly in the logic of the results.

4. In the discussion should be mentioned more the limitations of the study. There is no cell culture experimental model for PD!

Author Response

  1. There are minor scientific English issues that the authors should revise (e.g. Western blot should be written with capital letter always, etc.).

Reply: We thank the referee for his/her comment, we have revised minor scientific English correcting several typos.

  1. Figure legends (multiple) should be independent and clear for the reader. Where is RA coming from?

"Spectrophotometric analysis on the effects of PACAP on cell viability following MPP+ 308
treatment on dopaminergic neurons (RA)"?

Reply: We thank the referee for his/her comment, we add a paragraph 2.5 for the Differentiation of SH-SY5Y cells to explain better how the differentiation was conducted and we have modified (RA) in SH-SY5Y/RA. We have also separated Figure legends to make them clearer.

  1. If one carefully looks in Figure 5, there are many same cells marked both for apoptosis and necrosis. I do not see that clearly in the logic of the results.

Reply: We thank the referee for his/her comment, we have substitute Figure 6 and Figure 8 to make them clearer.

  1. In the discussion should be mentioned more the limitations of the study. There is no cell culture experimental model for PD!

Reply: We thank the referee for his/her comment, we have explained better in the Discussion section our PD in vitro model.

Reviewer 2 Report

Barra and colleagues investigated the neuroprotective effects of PACAP wrapped in gH625 decorated liposome in a model of PD consisting of MPP+ treated SH-SY5Y spheroids in a double chamber bioreactor mimicking blood-brain barrier(BBB). The authors relied on prestoblue and ROS assays to assess the neuroprotective effects of gH625-lipoPACAP against MPP+. 

This manuscript extensively introduced the SH-SY5Y with MPP+ as an in vitro model for PD, followed by a clear and organized methods section. However, the authors failed to character the aggregated cells, the liposome capsule, the LiveBox2 bioreactor, or the BBB functionality in the fluid dynamic model. The manuscript did not discuss or verify the apparent active transportation of gH625-lipoPACAP across the artificial BBB. More importantly, the statistics in the results were unclear and incomplete, making it challenging to support the conclusion. 

Specific points:

1. In the introduction, the authors detailed the SH-SY5Y in vitro model for PD. However, the authors omitted the limitation of the cell-line model. For example, as the authors stated in line 45, the SH-SY5Y cells are locked in an early neuronal stage, thus inherently lacking the hallmarks of the degeneration of the dopaminergic neurons in the human SNc in PD patients. 

2. In lines 71-73, the authors incorrectly cited a research article [reference 22] on brain trauma (hemorrhage) to support the idea of neuropeptide in regulation in PD.

3. The authors introduced the size classification of liposomes In line 91 and detailed liposome characterization in section 2.3. However, the manuscript did not include characterization results such as size distribution, loading percentage, or gH-625 coverage. 

4. Similarly, the authors briefly described the Livebox2 bioreactor and included a diagram in Figure 1. Yet, the report contains no picture or no description, such as the dimensions and permeability of the middle membrane. The organ-on-a-chip literature reported custom-made devices with great details. To replicate this research and compare it with published results, the authors need to document this adapted commercial bioreactor with all necessary information. 

5. In line 145, the authors need to specify how many freeze-thaw cycles are required: eight "to" ten times?

6. In the 3D SH-SY5Y in dynamic culture, the authors stated the cells were aggregated into 3D cultures. However, the authors did not characterize the size distribution or the number of cells per aggregate in the 100 µL/min condition, which is critical for culturing spheroids.

7. In lines 224-225, the authors seeded the mouse endothelial brain cells to mimic "a reliable blood-brain barrier(BBB)." The authors included tight junction protein analysis in the 3.5 western blots (Figures 11 and 12). However, the authors did not have basic BBB functional characterizations, such as TEER value and permeability test.

8. In Figures 1 and 2, the authors tested cell viabilities in conditions with MPP+ and MPP+ with PACAP treatment at different concentrations ranging from 0.5 to 1.5 mM. The figures were unclear, and the statistics were confusing. In both figures, the x-axises are unreadable. The dose-response of MPP+ is unclear. Its toxicity is not significant compared with RA control in Figure 2 and is not statistically significant in Figure 1. Moreover, the dose-response of PACAP with or without a liposome capsule is not presented in the data. Thus, the effectiveness of the gH625-lipoPACAP cannot be accessed. 

9. In Figure 4, the authors used selective images to demonstrate that PACAP receptors were presented in the 3D cells. Although the section and staining of the spheroids were technically challenging, the images were blurry and unable to illustrate the spatial distributions of the receptors. The authors fail to provide a quantitative analysis of the receptors or any in-depth discussion regarding the expression of the PACAP receptors.

10. The authors used Figures 5 to 8 to demonstrate that 1.5mM of MPP+ can induce apoptotic to necrotic shift in the spheroids, which has been reported in the literature.  Although the figures are informative, the authors provide a biased comparison: there are 60000 cells in Figure 6, whereas 300000 cells in Figure 7, merely one-fifth of cells were selected for the MPP+ test. 

11. In the spectrofluorometric analysis (Figure 9), the authors presented a result of a higher concentration of LC than UC,  indicating active transportation of gH625-lipoPACAP-Rho across the artificial BBB, given that passive crossing will result in equilibrium in both chambers at most. Unfortunately, the authors did not verify or discuss this observation in the manuscript. 

12. The authors did not provide a necessary introduction for the examined protein in the western blots. In addition, the authors did not compare or discuss the higher ZO-1 expression with other BBB systems in the literature or further characterize the BBB permeability. 

13. The statistic of quantitative measurement of ROS in Figure  13 is confusing. The authors need to clarify the statement that "compared to the spheroids treated only with MPP+ whose value is halved compared to the positive control(PC)." The PC should not be included in the comparison for evaluation of PACAP effectiveness. 

14. In lines 430-432, the authors need to define the term "restoration." Comparing PACAP/MPP+ with PACAP, how could MPP+ contribute to the higher apparent viability? What is the possible mechanism for PACAP restoring the MPP+ damage resulting in more viable cells?

Author Response

  1. In the introduction, the authors detailed the SH-SY5Yin vitromodel for PD. However, the authors omitted the limitation of the cell-line model. For example, as the authors stated in line 45, the SH-SY5Y cells are locked in an early neuronal stage, thus inherently lacking the hallmarks of the degeneration of the dopaminergic neurons in the human SNc in PD patients. 

Reply: We thank the referee for his/her comment. In the present manuscript we mentioned a limitation in the study of this undifferentiated cell line for PD adding that the help of treatments like retinoic acid (RA), can force this cell line to differentiate into dopaminergic neurons and be used for the treatment of PD. In particular, in our previous work on Frontiers in Physiology (19th August 2022, doi 10.3389/fphys.2022.932099, ref.18) we have made Immunofluorescence assay for neural marker for SH-SY5Y differentiated with RA demonstrating that the help of RA led to the DAergic formation. However, more references have been added in the Introduction section, as to why this undifferentiated cell line has limitations for PD.

  1. In lines 71-73, the authors incorrectly cited a research article [reference 22] on brain trauma (hemorrhage) to support the idea of neuropeptide in regulation in PD.

Reply: We thank the referee for his/her comment, we have substituted that reference.

  1. The authors introduced the size classification of liposomes In line 91 and detailed liposome characterization in section 2.3. However, the manuscript did not include characterization results such as size distribution, loading percentage, or gH-625 coverage. 

Reply: We thank the referee for his/her comment, we add the paragraph 3.1 for liposome characterization.

  1. Similarly, the authors briefly described the Livebox2 bioreactor and included a diagram in Figure 1. Yet, the report contains no picture or no description, such as the dimensions and permeability of the middle membrane. The organ-on-a-chip literature reported custom-made devices with great details. To replicate this research and compare it with published results, the authors need to document this adapted commercial bioreactor with all necessary information. 

Reply: We thank the referee for his/her comment, we add the information requested in the Introduction section and in the paragraph 2.10 Spectrofluorimetry assay. We have added a picture Figure 1a to describe better the millifluidic bioreactor. We have recently published an article on Frontiers in Physiology (19th August 2022, doi: 10.3389/fphys.2022.932099 ref. 18) in which we explain the difference among our millifluidic systems, and the bioreactor used in this work. 

  1. In line 145, the authors need to specify how many freeze-thaw cycles are required: eight "to" ten times?

Reply: We thank the referee for his/her comment, we have modified in the manuscript.

  1. In the 3D SH-SY5Y in dynamic culture, the authors stated the cells were aggregated into 3D cultures. However, the authors did not characterize the size distribution or the number of cells per aggregate in the 100 µL/min condition, which is critical for culturing spheroids.

Reply: We thank the referee for his/her comment, we have added the number of cells per aggregate in the paragraph 2.7. 3D SH-SY5Y/RA in dynamic culture.

  1. In lines 224-225, the authors seeded the mouse endothelial brain cells to mimic "a reliable blood-brain barrier(BBB)." The authors included tight junction protein analysis in the 3.5 western blots (Figures 11 and 12). However, the authors did not have basic BBB functional characterizations, such as TEER value and permeability test.

Reply: We thank the referee for his/her comment, we have performed permeability test and different assays to determine the presence of reliable blood-brain barrier in our recent article (doi: 10.3389/fphys.2022.932099 re.19). We have added the information requested in the paragraph 2.11 Protein extraction and Western Blot.

  1. In Figures 1 and 2, the authors tested cell viabilities in conditions with MPP+ and MPP+ with PACAP treatment at different concentrations ranging from 0.5 to 1.5 mM. The figures were unclear, and the statistics were confusing. In both figures, the x-axises are unreadable. The dose-response of MPP+ is unclear. Its toxicity is not significant compared with RA control in Figure 2 and is not statistically significant in Figure 1. Moreover, the dose-response of PACAP with or without a liposome capsule is not presented in the data. Thus, the effectiveness of the gH625-lipoPACAP cannot be accessed. 

Reply: We thank the referee for his/her comment, we have substitute Figure 1 and Figure 2 separating them to make more readable.

  1. In Figure 4, the authors used selective images to demonstrate that PACAP receptors were presented in the 3D cells. Although the section and staining of the spheroids were technically challenging, the images were blurry and unable to illustrate the spatial distributions of the receptors. The authors fail to provide a quantitative analysis of the receptors or any in-depth discussion regarding the expression of the PACAP receptors.

Reply: We thank the referee for his/her comment, we have improved images.  Our intention within the manuscript was not to provide a quantitative analysis of the receptors, rather to observe their presence within our 3D SH-SY5Y / RA as, in case of negative result, to evaluate the PACAP action on spheroids would have been more difficult. Within the manuscript, however, we have added references regarding the presence of receptors in this cell line, also adding a part in the Introduction section.

  1. The authors used Figures 5 to 8 to demonstrate that 1.5mM of MPP+ can induce apoptotic to necrotic shift in the spheroids, which has been reported in the literature.  Although the figures are informative, the authors provide a biased comparison: there are 60000 cells in Figure 6, whereas 300000 cells in Figure 7, merely one-fifth of cells were selected for the MPP+ test. 

Reply: We thank the referee for his/her comment, we have substituted Figure 6 and Figure 8 normalizing to 200.000 cells.

  1. In the spectrofluorometric analysis (Figure 9), the authors presented a result of a higher concentration of LC than UC,  indicating active transportation of gH625-lipoPACAP-Rho across the artificial BBB, given that passive crossing will result in equilibrium in both chambers at most. Unfortunately, the authors did not verify or discuss this observation in the manuscript. 

Reply: We thank the referee for his/her comment, we have discussed it in the Discussion section.

  1. The authors did not provide a necessary introduction for the examined protein in the western blots. In addition, the authors did not compare or discuss the higher ZO-1 expression with other BBB systems in the literature or further characterize the BBB permeability. 

Reply: We thank the referee for his/her comment, we have discussed in the paragraph 3.6 Western Blot

  1. The statistic of quantitative measurement of ROS in Figure  13 is confusing. The authors need to clarify the statement that "compared to the spheroids treated only with MPP+ whose value is halved compared to the positive control(PC)." The PC should not be included in the comparison for evaluation of PACAP effectiveness. Reply: We thank the referee for his/her comment. We updated figure 13 and its figure legend. Experimental groups are now compared against MPP+. PC is reported just to consider the maximum ROS.  
  2. In lines 430-432, the authors need to define the term "restoration." Comparing PACAP/MPP+ with PACAP, how could MPP+ contribute to the higher apparent viability? What is the possible mechanism for PACAP restoring the MPP+ damage resulting in more viable cells?

Reply: We thank the referee for his/her comment. We have discussed in Discussion section

Round 2

Reviewer 2 Report

The authors have made significant improvements in the revised manuscript. They have addressed most of the concerns listed in the last review. In this revision, the authors add more data regarding the characterization of the aggregated cells and LiveBox2 bioreactor.  However, two main points have not been sufficiently resolved: 

1. The new figures 2 and 3 were much more precise and easier to understand. The dose response of MPP+ remains unclear and counterintuitive. In Figure 2, the PACAP was more protective against a higher concentration of MPP+ than a lower one (1.5 mM vs. 1 mM). In addition, in Figure 3, the lower concentration of PACAP was more protective than the higher concentration against 1mM MPP+ (10-8 M vs. 10-7 M).  Also, the order of panels in Figure 3 is not from low to high MPP+.  Thus, it is challenging to conclude that "an increase in viability at 10-7 M and 10-8M (line 476).

2. Lack of negative controls in figures 13 and 14. In Figure 13, the authors did not provide the baseline of ROS in untreated conditions. More importantly, in Figure 14, the 3D cultures treated with gH625-lipoPACAP-Rho showed much lower absorbance at 570nm compared with RA controls in Figures 2 and 3. Without providing the baseline of the untreated 3D cultures, the comparison is incomplete; and it is insufficient to conclude that there was "a restoration by PACAP on 3D SH-SY5Y/RA when treated with MPP+." (lines 628-629)

Author Response

Dear Biomedicine Editor,

We thank the referee for the other comments to the manuscript.

We hope that the present version of the manuscript is suitable for the publication.

Sincerely,

Salvatore Valiante

  1. The new figures 2 and 3 were much more precise and easier to understand. The dose response of MPP+remains unclear and counterintuitive. In Figure 2, the PACAP was more protective against a higher concentration of MPP+than a lower one (1.5 mM vs. 1 mM). In addition, in Figure 3, the lower concentration of PACAP was more protective than the higher concentration against 1mM MPP+ (10-8 M vs. 10-7 M).  Also, the order of panels in Figure 3 is not from low to high MPP+.  Thus, it is challenging to conclude that "an increase in viability at 10-7 M and 10-8M (line 476).

Reply: We have redone the statistical analysis for Figures 2 and 3. In figure 2 the values ​​for the PACAP are significant by comparing them to the appropriate control (MPP+). In figure 3, gH625lipoPACAP-Rho is significant respect to the MPP+ at concentrations of 10-6 M and 10-8 M. It is of note that cell viability for 10-7M PACAP experimental class is always higher than that of control (MPP+), although the increase results statistically significant only at 1mM MPP+. Similarly, 10-8M PACAP always increases cell viability but at 0,5mM MPP+ where the result has P value higher than 0.05. Although this could undermine the statistical power of the analysis (the reason of this remains to be elucidated but it can take into account several reasons e.g. PACAP receptor different binding affinity for PACAP or slightly different physiological conditions in experiments, as suggested by the different values of RA between fig 2 and 3), submitted data yet support the neuroprotective role of PACAP along with other provided here and published data. Hence, we can confidently state that PACAP, released by our nanodelivery tool, has consistent biologically activity. However, we changed the sentence "an increase in viability at 10-7 M and 10-8M", avoiding generalization and specifying deeper in details figs 2 and 3. There was a simple typo “0.05mM” of MPP+ instead of “0.5 mM” MPP+; we have fixed up the error in Figure 3.

  1. Lack of negative controls in figures 13 and 14. In Figure 13, the authors did not provide the baseline of ROS in untreated conditions. More importantly, in Figure 14, the 3D cultures treated with gH625-lipoPACAP-Rho showed much lower absorbance at 570nm compared with RA controls in Figures 2 and 3. Without providing the baseline of the untreated 3D cultures, the comparison is incomplete; and it is insufficient to conclude that there was "a restoration by PACAP on 3D SH-SY5Y/RA when treated with MPP+." (lines 628-629)

Reply: We thank the referee for his/her comment, for DCFH2-DA assay,

the values of the untreated cells were calculated as baseline and subtracted from the treated group values. We have now reported this information in the manuscript. Prestoblue assay was made to compare 3D cells treated with MPP+ in simultaneous or subsequent treatment with gH625-liposomePACAP-Rho; for this reason, we have not provided negative control in the text. Furthermore, it must consider that the first Prestoblue (Materials and Methods section 2.6) was performed on cells seeded in static conditions while the second Prestoblue (Materials and Methods section 2.13) assay was made on 3D spheroid in fluid-dynamic conditions. Although the absorbance values are normalized, it is to consider that the experimental conditions are very different, make challenging the direct comparison of two histograms belonging to two so different experimental conditions (e.g., different exposition time, number of cells, different architecture, nutrient availability etc). Moreover, as reported in different references, fluorescence or absorbance results can be influenced from cells number based on Prestoblue concentration: a smaller number of cells can have higher absorbance/fluorescence values ​​compared to a high number of cells as happens in our 3D SH-SY5Y/RA data. This test can offer a high sensitivity for long incubation time, but values are higher in small population with poor impaired metabolism (Sonnaert M, Papantoniou I, Luyten FP, Schrooten JI. Quantitative Validation of the Presto Blue Metabolic Assay for Online Monitoring of Cell Proliferation in a 3D Perfusion Bioreactor System. Tissue Eng Part C Methods. 2015 Jun;21(6):519-29. doi: 10.1089/ten.TEC.2014.0255. Epub 2015 Mar 31. PMID: 25336207; PMCID: PMC4442584; Luzak B, Siarkiewicz P, Boncler M. An evaluation of a new high-sensitivity PrestoBlue assay for measuring cell viability and drug cytotoxicity using EA.hy926 endothelial cells. Toxicol In Vitro. 2022 Sep;83:105407. doi: 10.1016/j.tiv.2022.105407. Epub 2022 Jun 1. PMID: 35659575.).

Round 3

Reviewer 2 Report

The authors have made sufficient revisions to the manuscript to address the last two concerns in the previous review.  The refined statistics are more precise and easier to follow.  The issue with the lack of negative controls in figures 13 and 14 is resolved by the explanation in the reply.  The manuscript is now recommended for publication.